

# Melatonin improves the efficiency of super-ovulation and timed artificial insemination in sheep

Yukun Song[1], Hao Wu[2], Xuguang Wang[1], Aerman Haire[1], Xiaosheng Zhang[3], Jinlong Zhang[3], Yingjie Wu[2], Zhengxing Lian[2], Juncai Fu[2], Guoshi Liu[2] and Abulizi Wusiman[1]

[1] College of Animal Science, Xinjiang Agricultural University, Urumqi, Xinjiang, China

[2] National Engineering Laboratory for Animal Breeding, Key Laboratory of Animal Genetics and Breeding of the Ministry of Agriculture, Beijing Key Laboratory for Animal Genetic Improvement, College of Animal Science and Technology, China Agricultural University, Beijing, Beijing, China

[3] Institute of Animal Husbandry and Veterinary, Academy of Agricultural Sciences of Tianjin, Tianjin, Tianjin, China

## ABSTRACT

It has been well proved that melatonin participates in the regulation of the seasonal reproduction of ewes. However, the effects of short term treatment of melatonin on ewe's ovulation are still to be clarified. In this study, the effects of melatonin on the number of embryos harvested from superovulation, and the pregnant rate in recipients after embryo transferred have been investigated. Hu sheep with synchronous estrus treatment were given melatonin subcutaneously injection (0, 5, and 10 mg/ewe, respectively). It was found that the estrogen level in the group of 5 mg melatonin was significantly higher than that of other two groups at the time of sperm insemination ($p < 0.05$). The pregnant rate and number of lambs in the group of 5 mg melatonin treatment was also significantly higher than that of the rests of the groups ($P < 0.05$). In another study, 31 Suffolk ewes as donors and 103 small-tailed han sheep ewes as recipients were used to produce pronuclear embryo and embryo transfer. Melatonin (5 mg) was given to the donors during estrus. The results showed that, the number of pronuclear embryos and the pregnancy rate were also significantly higher in melatonin group than that in the control group. In addition, 28 donors and 44 recipient ewes were used to produce morula/blastocyst and embryo transferring. Melatonin (5 mg) was given during estrus. The total number of embryos harvested ($7.40 \pm 1.25$/ewe vs. $3.96 \pm 0.73$/ewe, $P < 0.05$) and the pregnant rate ($72.3 \pm 4.6\%$ vs. $54.7 \pm 4.0\%$, $P < 0.05$) and number of lambs were also increased in melatonin group compared to the control group. Collectively, the results have suggested that melatonin treatment 36 hours after CIDR withdrawal could promote the number and quality of embryos *in vivo* condition and increased the pregnant rate and number of lambs.

Corresponding authors
Guoshi Liu, gshliu@cau.edu.cn
Abulizi Wusiman, abulizi68@126.com

## INTRODUCTION

The estrous cycles of most sheep are regulated by the switch of seasons. With the reduced photoperiod and increased melatonin level, ewes adjust GnRH level and estrus cycles

(*Ortavant et al., 1988*). Thus, as a photoperiodic signal molecule melatonin regulates the reproductive activity of ewes. Melatonin is also a potent antioxidant and free radical scavenger (*Tan et al., 1993*). Under the consideration, melatonin can also protect the reproductive tissues and organs. A various of ROS (reactive oxygen species such as $OH^{\cdot-}$ and $O_2^-$) will damage the DNA and the lipid of cell membrane, accelerate apoptosis in reproductive system (*Agarwal, Gupta & Sharma, 2005*). For example, oxidative stress induces two cell developmental block, apoptosis and infertility (*Tatone & Amicarelli, 2013*; *Devine, Perreault & Luderer, 2012*). The reason is that oxidative stress reduces the quality of oocytes which is an important factor for sheep fertility. Melatonin scavenges $OH^-$, $H_2O_2$ and other reactive oxygen species; therefore, it contributes to effectively reduce oxidative DNA damage and cell apoptosis during ovulation (*Miyamoto et al., 2010*). Melatonin also reduces oxidative damage to mitochondrial DNA (*Galano, Tan & Reiter, 2018*). It was reported that melatonin preserved the normal distribution of mitochondria, mitochondrial DNA copy number membrane potential (MMP), and ATP level (*Liang et al., 2017*). Melatonin improves the quality of bovine oocyte, oocyte maturation, efficiency of *in vitro* fertilization and embryo development (*He et al., 2016b*). Melatonin administration improved the conception rates in mice and cows (*Guo et al., 2017*). Melatonin added during culture of mouse prokaryotic embryos significantly increased blastocyst rate, pregnancy rate after transplantation, average number of offspring and survival rate of offspring (*Yang et al., 2017b*). For deer, melatonin subcutaneous implantation also improved the quantity and quality of super ovulatory oocytes (*He et al., 2016a*). Therefore, melatonin is probably the key factor to improve fecundity by improving the quality of oocytes in mammals.

Melatonin also plays an important role in the establishment and maintenance of pregnancy in animals. In mouse, extremely high expression of melatonin receptor 1 (MT1) was observed in granulosa cells after human chorionic gonadotropin (hCG) treatment (*He et al., 2016a*). At the same time, melatonin synthetic enzyme in cumulus cells was upregulated by hCG injection and high level of melatonin in follicle fluid was detected (*Tian et al., 2017b*) and this was observed also in porcine follicle fluid (*Shi et al., 2009*). Further study demonstrated that melatonin and its receptor MT1 regulated the downstream signaling pathway of hCG (LH) including the luteinization of granulosa cells in mice (*He et al., 2016a*). The deletion of MT1 receptor severely impairs the fertility of mice due to reduced oocyte number and quality (*Zhang et al., in press*).

In addition, melatonin subcutaneous implantation in sheep increase the number of corpus luteum and pregnancy rate (*Zhang et al., 2013*). However, there is no report regarding the effect of melatonin short term administration during estrus on the reproductive efficiency and embryo production in animals, particularly in ewes. Therefore, this study was conducted to test whether melatonin short term administration can improve the synchronized estrus pregnant rate and the production of both pronuclear and morula/blastocyst embryos in ewes.

## MATERIAL AND METHODS

### Chemicals

CIDRs (Controlled Internal Drug Release) which contains 300 mg progesterone were purchased from Pfizer Animal Health (New Zealand). Follicle stimulating hormone (FSH), luteinizing hormone (LH) and pregnant mare serum gonadotropin (PMSG) were from Ningbo Sansheng Pharmaceutical Industry Co., Ltd. (Zhejiang Sheng, China). Melatonin and all other reagents, unless specified, were purchased from Sigma-Aldrich Co. (St. Louis, MO, USA).

### Animals

Female sheep (Hu, Suffolk and Small-tailed Han) from Aoxin Animal Husbandry (Beijing, China) and Zhenxin Farmers's Professional Association Organization (Uygur Autonomous Region, China), at the age of 2–4 years old with normal reproductive cycle, healthy and generally the similar body weight were selected for the experiments. All experimental protocols concerning the handling of animals were performed in accordance with the requirements of the Institutional Animal Care and Use Committee at the Xinjiang Agricultural University (permission number: 2017003).

### Experiment design

#### Experiment 1

Fifty-seven ewes were randomly divided into three groups and then treated with CIDRs to induce synchronized estrus. All the ewes assess freely to the same feed and drink water. The CIDRs were removed 13 days later, blood samples were collected and the ewes were injected with PMSG. The second blood collection was conducted 36 h after the CIDR removal and melatonin (0, 5, 10 mg), was subcutaneously injected at the same time, respectively. Artificial insemination was conducted 48 h after the removal of the CIDR, and the third blood collection was conducted. B-ultrasound examination was conducted 45 days after artificial insemination and ewes with pregnancy were recoded.

#### Experiment 2

Thirty-one Suffolk ewes as donors and 103 Small-tailed Han ewes as recipients were selected, and the donors were divided into two groups and underwent the stimulated ovulation procedure. Among the donors, 13 ewes were subcutaneously injected with 5mg melatonin 36 h after the CIDR was removed and 18 ewes served as control group, and LH was injected to all the donors at the same time. Then the donors were inseminated by laparoscope, and 10 h later pronuclear embryos were surgically collected, and the pronuclear embryos were observed by stereomicroscope and selected depends on the morphology. The number of corpora luteal (CL) and embryos harvested were recorded. The embryos with normal morphology were transferred into the oviduct of recipients. The examination of pregnancy was performed using B-ultrasound 45 days after embryo transfer.

#### Experiment 3

Twenty-eight Suffolk ewes as donors and 44 Small-tailed Han ewes as recipient were selected and divided into two groups randomizedly. The donor ewes were treated to induce

superovulation, and 36 h after CIDR removal LH and melatonin (5 mg) was subcutaneously injected into donors at the same time. Six days after artificial insemination by laparoscopic, morula or blastocyst were surgically harvested from the uterine horn of the donor, and the number of CL and embryos were recorded. The recipient was surgically operated with the laparoscopic surgery to find the uterine horn, finally the embryo was transplanted into the uterine horn from the 5 cm junction between the uterus and the fallopian tube by the the transplantation pipette. The examination of pregnancy was performed using B-ultrasound 45 days after embryo transfer.

## Semen preparation

In this study, fresh semen was collected from male sheep of black Suffolk using a vaginal prosthesis, and then the semen was diluted and store at 36 °C. Fresh semen is injected into the uterine horn when its vitality reaches 0.6 or more by laparoscopic insemination.

## Blood sample collection and hormones analyze

For Experiment 1, 5 ml blood samples was collected and stored in −80 °C freezer for measurement. Hormones level were determined was conducted by Beijing North Biotechnology Research Institute by radioimmunoassay.

## Statistical analysis

All data were presented as means ± SEM. The data were analyzed using ANOVA and followed by LSD and Duncan tests for the differences between treatments (SPSS software; SPSS, Inc., Chicago, IL, USA); $P < 0.05$ was used as the criterion for the significance of the difference.

# RESULTS

## Effects of melatonin injection on hormones

Firstly, the effects of melatonin on FSH, LH and E2 were evaluated at three different time points (Table 1). At the time of artificial insemination, the melatonin in the serum of ewes in 5 mg group (509.0 ± 67.5 pg/ml) was significantly higher than that of the other control group (330.2 ± 38.7 pg/ml) ($p < 0.05$). But there was no difference in melatonin levels between the three groups at the time of CIDR withdrawal and estrus. The FSH level in the control group was always in a high level, but the increase of FSH from estrus to insemination (0.2 mIU/ml) in the 5 mg group was significantly higher than that in the other two groups (0.03 mIU/ml and 0.09 mIU/ml, $p < 0.05$). LH level was increased from CIDR withdrawal to estrus, and gradually decreased after the peak level. LH level in the 5 mg group (4.59 ± 0.4 mIU/ml) was significantly higher than that in the other two groups (4.02 ± 0.3 mIU/ml, 4.27 ± 0.3 mIU/ml) at the time of insemination ($p < 0.05$). Progesterone concentration in the 10 mg group was the highest, followed by a downward trend, significantly higher than that in the control group at the time of withdrawal CIDR and estrus ($p < 0.05$), and significantly higher than 5 mg group at the time of estrus ($p < 0.05$). The concentration of progesterone of the 10 mg treated group was highest at the time of insemination, but there was no significant difference between the two groups ($p > 0.05$). There was no difference

**Table 1  Effects of melatonin injection on FSH, LH, P$_4$ and E$_2$.**

| Time | Group | MT (pg/ml) | FSH (mIU/ml) | LH (mIU/ml) | P$_4$(mIU/ml) | E$_2$ (mIU/ml) |
|---|---|---|---|---|---|---|
| | 5 mg | 363.59 ± 63.87[a] | 2.04 ± 0.08[a] | 4.93 ± 0.25[a] | 0.26 ± 0.04[ab] | 14.61 ± 2.56[a] |
| Withdrew CIDR | 10 mg | 390.69 ± 64.22[a] | 2.05 ± 0.07[a] | 4.93 ± 0.34[a] | 0.30 ± 0.05[a] | 15.06 ± 1.30[a] |
| | Control | 350.95 ± 63.16[a] | 2.16 ± 0.14[a] | 4.64 ± 0.29[a] | 0.21 ± 0.02[b] | 15.04 ± 1.39[a] |
| | 5 mg | 458.69 ± 48.40[a] | 1.86 ± 0.09[b] | 5.39 ± 0.52[a] | 0.17 ± 0.02[b] | 23.11 ± 3.62[a] |
| Estrus | 10 mg | 458.09 ± 60.60[a] | 1.79 ± 0.13[b] | 5.41 ± 0.49[a] | 0.22 ± 0.02[a] | 24.49 ± 2.50[a] |
| | Control | 393.37 ± 51.53[a] | 2.05 ± 0.15[a] | 5.26 ± 0.48[a] | 0.16 ± 0.02[b] | 22.96 ± 2.01[a] |
| | 5 mg | 509.00 ± 67.52[a] | 2.06 ± 0.13[a] | 4.59 ± 0.38[a] | 0.14 ± 0.02[a] | 19.23 ± 2.66[a] |
| Insemination | 10 mg | 457.24 ± 40.65[ab] | 1.76 ± 0.14[b] | 4.02 ± 0.27[b] | 0.18 ± 0.03[a] | 13.78 ± 1.77[b] |
| | Control | 330.23 ± 38.72[b] | 2.14 ± 0.26[a] | 4.27 ± 0.31[b] | 0.14 ± 0.02[a] | 14.07 ± 2.86[b] |

Notes.
Different letters in the same column at same time point indicate significant difference ($p < 0.05$).

**Table 2  Effect of melatonin treatment on pregnancy of ewes.**

| Group | Ewes | Pregnant ewes | Pregnancy rate | Lambs/ewes (%) |
|---|---|---|---|---|
| 5 mg | 21 | 14 | 66.67 ± 4.76[a] | 40/14 (285.7%)[a] |
| 10 mg | 20 | 8 | 40.48 ± 6.30[b] | 22/8 (275.0%)[a] |
| Control | 18 | 7 | 37.62 ± 5.78[b] | 22/7 (314.3%)[a] |

Notes.
Different letters in the same column indicate significant difference ($p < 0.05$).

in estradiol level between the two groups at the time of CIDR withdrawal and estrus. At the time of insemination, 5 mg group (19.2 ± 2.7 pg/ml) was significantly higher than the other two groups (13.8 ± 1.8 pg/ml, 14.1 ± 2.9 pg/ml, $p < 0.05$).

## Effects of melatonin on the pregnancy and number of lambs born in sheep

Melatonin was subcutaneously injected to the ewes 36 h after the withdrawal of CIDRs, at the dosage of 5 mg (21 ewes), 10 mg (20 ewes), and control group (18 ewes). Then the ewes received artificial insemination and the pregnancy was examined 45 days later. As shown in Table 2, the pregnancy rate of ewes with melatonin 5 mg injection (66.67 ± 4.76%) was significantly higher than that of the other two groups, respectively (40.48 ± 6.30%, 37.62 ± 5.78%, $p < 0.05$), and there was no significant difference between the 10 mg group and the control group. As for the number of lambs born , there was no difference among all three groups ($p > 0.05$).

## Effects of melatonin on embryo production and pregnancy rate in sheep

To know how melatonin affects sheep reproductive activity, the pronuclear embryos and blastocysts of the donors were harvested and then transferred to recipients. It was found that melatonin injection slightly increased the number of CL (8.78 ± 1.78 vs 8.44 ± 1.13) and pronuclear embryos (8.8 ± 1.9/ewe vs 8.3 ± 1.0/ewe), ($p > 0.05$) (Table 3). However, the pregnancy rate of embryos and birth rate of lambs in melatonin 5 mg group were significantly higher than that of the control group (43.3 ± 6.1% vs 25.3 ± 4.9%, and

**Table 3  Effect of different treatments on pronuclear embryo production after superovulation of donor ewes.**

| Group | Donors | Corpus Luteum | Embryos | Normal Embryos |
|---|---|---|---|---|
| 5 mg | 13 | $8.78 \pm 1.78$[a] | $8.81 \pm 1.86$[a] | $8.81 \pm 1.86$[a] |
| Control | 18 | $8.44 \pm 1.13$[a] | $8.31 \pm 1.00$[a] | $8.31 \pm 1.00$[a] |

Notes.
Different letters in the same column indicate significant difference ($p < 0.05$).

**Table 4  Effect of different treatments on pregnancy rate after pronuclear embryo transplantation in recipient sheep and lambs born.**

| Group | Recipient | Embryos transferred | Pregnancy rate | Lambs/ewes |
|---|---|---|---|---|
| 5 mg | 45 | $2.5 \pm 0.2$ (115/45) | $43.3 \pm 6.1\%$ (20/45)[a] | 24/45 ($54.0 \pm 4.0\%$)[a] |
| Control | 58 | $2.6 \pm 0.1$ (150/58) | $25.3 \pm 4.9\%$ (14/58)[b] | 15/58 ($27.7 \pm 9.0\%$)[b] |

Notes.
Different letters in the same column indicate significant difference ($p < 0.05$).

**Table 5  Effect of different treatments on morula/blastocyst production after superovulation of donor.**

| | Donors | Average luteum | Average embryos | Normal embryos |
|---|---|---|---|---|
| 5 mg | 15 | $10.33 \pm 1.37$[a] | $7.80 \pm 1.25$[a] | $7.40 \pm 1.25$[a] |
| Control | 13 | $8.08 \pm 1.13$[a] | $4.08 \pm 0.70$[b] | $3.96 \pm 0.73$[b] |

Notes.
Different letters in the same column indicate significant difference ($p < 0.05$).

$54.0 \pm 4.0\%$ vs $27.7 \pm 9.0\%$ $p < 0.05$) (Table 4). During the period of estrus, the donor who received melatonin treatment significantly increased the total number of morula embryo/blastocyst at 6 days after insemination ($7.4 \pm 1.3$/ewe, $4.0 \pm 0.7$/ewe, $p < 0.05$) (Table 5), and also the pregnancy rate was significantly increased after embryo transfer compared to the control group ($72.3 \pm 4.6\%$ vs $54.7 \pm 4.0\%$, $p < 0.05$), for the number of lambs the ewes born, there was no difference among two groups ($p > 0.05$) (Table 6).

## DISCUSSION

Current study has demonstrated that short term melatonin treatment during estrus elevates estradiol and LH levels, improves oocyte quality and leads to the increase of the pregnant rate in ewes. These observations are consistent with previous studies which indicates that melatonin treatment increases serum LH (*Zarazaga et al., 2009*) and progesterone levels in sheep (*Abecia, Forcada & Zuniga, 2002*). *Wang et al. (2014)* found that the melatonin level decreased after the removal of CIDR in deer and this is similar to our observation in the current study. We also observed that melatonin administration significantly increased serum melatonin level, it enhanced the LH and progesterone levels in turn, subsequently led to increase in the numbers of embryos and better pregnant rate.

Ovulation is similar to inflammatory reaction that produces a large amount of ROS and reactive nitrogen (RNS) (*Gupta et al., 2006*). In the early pregnancy, ROS inhibits

**Table 6 Effect of different treatments on pregnancy rate after morula/blastocyst transplantation in recipient sheep.**

| Group | Recipient | Embryos transferred | Pregnancy rate | Lambs/ewes |
|---|---|---|---|---|
| 5 mg | 25 | $1.88 \pm 0.13$ (47/25)[a] | $72.3 \pm 4.6\%$ (18/25)[a] | 19/18 ($105.7 \pm 10.0\%$)[a] |
| Control | 22 | $1.86 \pm 0.15$ (41/22)[a] | $54.7 \pm 4.0\%$(12/22)[b] | 14/12 ($116.7 \pm 14.4\%$)[a] |

Notes.
Different letters in the same column indicate significant difference ($p < 0.05$).

progesterone production from corpus luteal cells causes luteal CL regression (*Al-Gubory et al., 2012*), and also induces ovarian cell apoptosis (*Korzekwa et al., 2006*). As a potent antioxidant, melatonin detoxifies ROS including $OH^-$, $O^{2-}$ and $H_2O_2$ and reduces the oxidative damage of ovarian cells (*Loren et al., 2017*) and thus, improves the fertility and fecundity of sheep by improving the survival rate of corpus luteum and embryos (*Tamura et al., 2014*). Numerous studies have proved that melatonin promotes the development of oocytes and embryos in sheep (*Tian et al., 2017a*), pigs (*Wang et al., 2017*), cattle (*Rodrigues-Cunha et al., 2016*), mice (*Zhang et al., 2016*) and humans (*Carlomagno et al., 2011*) *in vitro* by scavenging ROS.

*Luridiana et al. (2015)* reported that the melatonin implantation in ewe at the age of 5-6 with 3.5–4.0 body condition score (BSC) in spring improved fertility of ewes. In our study, the pregnant rate of ewes received single melatonin injection was significantly increased as well as both pronuclear embryos and blastocysts compared with the control group. *Yang et al. (2017a)* observed that the melatonin injection before mating improved the pregnancy rate of Holstein cows preluded with elevated serum melatonin and progesterone levels. In our experiment, the progesterone level in serum did not change after melatonin treatment, but the estradiol was increased. Melatonin may benefit follicular development, and then increases estradiol synthesis to promote ovulation in female sheep.

In this study, we observed the effects of melatonin to synchronize estrous in ewe, especially in the donor ewes to benefit the embryo transplantation. Short term of melatonin treatment during estrus might improve the uterine environment of ewe and significantly increased the pregnant rate. In addition, melatonin improves the quality and quantity of embryos and this may also contribute to the increased pregnant rate. These data provide strong support for the application of melatonin in sheep to improve the reproductive outcome industrially.

## CONCLUSION

Melatonin at 5 mg subcutaneously injected into the neck during estrus would increase the level of melatonin and estradiol in the blood of ewe, and melatonin promoted embryo production and the pregnancy rate in the awes naturally mated or embryo transfer. Meanwhile, melatonin had beneficial effects on recipients for embryo transfer. Altogether, melatonin could be used to the improve the number of lambs.

### Funding

This research is supported by a major project to cultivate new genetically modified organisms (2018ZX0800801B, 2016ZX08008003); Xinjiang uygur autonomous region science and technology support Xinjiang project (2016E02037). The funders had no role in study design, data collection and analysis, decision to publish, or preparation of the manuscript.

### Grant Disclosures

The following grant information was disclosed by the authors:
Major project: 2018ZX0800801B, 2016ZX08008003.
Xinjiang uygur autonomous region science and technology: 2016E02037.

### Competing Interests

The authors declare there are no competing interests.

### Author Contributions

- Yukun Song and Hao Wu conceived and designed the experiments, performed the experiments, analyzed the data, prepared figures and/or tables, approved the final draft.
- Xuguang Wang, Aerman Haire, Xiaosheng Zhang and Jinlong Zhang performed the experiments, approved the final draft.
- Yingjie Wu, Zhengxing Lian and Juncai Fu performed the experiments, contributed reagents/materials/analysis tools, approved the final draft.
- Guoshi Liu conceived and designed the experiments, performed the experiments, contributed reagents/materials/analysis tools, authored or reviewed drafts of the paper, approved the final draft.
- Abulizi Wusiman conceived and designed the experiments, performed the experiments, analyzed the data, prepared figures and/or tables, authored or reviewed drafts of the paper, approved the final draft.

### Field Study Permissions

The following information was supplied relating to field study approvals (i.e., approving body and any reference numbers):

Xinjiang Agricultural University approved this study (permission number: 2017003).

### Data Availability

The raw measurements are available in Tables S1 and S2.

### Supplemental Information

Supplemental information for this article can be found online at http://dx.doi.org/10.7717/peerj.6750#supplemental-information.

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
