# Peer review of "Melatonin improves the efficiency of super-ovulation and timed artificial insemination in sheep"

_PeerJ, doi:10.7717/peerj.6750_

## Round 0.1 · original submission · Minor Revisions

Please revised your paper based on the comments of all reviewers point by point.

Reviewer 1 ·

Basic reporting

no comment

Experimental design

no comment

Validity of the findings

no comment

Additional comments

This study provides data showing that short term of melatonin treatment during estrus improve the uterine environment of ewe and significantly increased the pregnant rate. Data also demonstrated that melatonin administration improves the quality and quantity of embryos and this may also contribute to the increased pregnant rate. The findings are interesting and provide strong support for the application of melatonin in sheep to improve the reproductive outcome industrially. However, the paper is poorly written and needs significant English editing for publication. It is recommended that the authors find an expert to correct the numerous English errors throughout the manuscript.

Reviewer 2 ·

Basic reporting

no comment

Experimental design

no comment

Validity of the findings

no comment

Additional comments

In the manuscript "Melatonin improves the efficiency of superovulation and timed artificial insemination in sheep (#34096) " by Yukun Song et al. The authors documented that the short-term effects of melatonin on ewe’s ovulation and the results showed that melatonin treatment at 36 hours after withdrawal CIDR could promote the number and quality of embryos in vivo and increased the pregnant rate and number of lambs. These results provide a support for the application of melatonin in sheep to improve the reproductive outcome industrially.
I have some minor comments that should be addressed as following:
1. How long does a half-life of melatonin have in the ewe blood?
2. How was the injected doses of melatonin determined?
3. What means of 0, 5, 10 mg were? For example, 5 mg is 5 mg melatonin per ewe or is 5 mg melatonin per kg bodyweight of ewe?
4. In line 130, “the embryos with normal morphology were transferred into the oviduct of recipients”. Please elaborate the specific process.
5. In line 139-142, please state how to prepare semen in vitro artificial insemination? For example, semen conservation, sperm capacitation, and so on.

6. In line154-155, why was the melatonin concentration higher in the serum of ewes in the 5 mg group, but not that of the 10 mg group?

·

Basic reporting

'no comment'

Experimental design

'no comment'

Validity of the findings

'no comment'

Additional comments

The aim of this paper was to investigate the role of Melatonin in superovulation and timed artificial insemination in sheep. The Authors found that melatonin treatment at 36 hours after withdrawal CIDR could promote the number and quality of embryos in the in vivo condition and increased the pregnant rate and number of lambs. The article is original and the Authors purpose is interesting, however the authors must to prepare thorough correction of the whole manuscript. In the result part, why the high concentration group had higher P4 in Estrus stage and lower pregnancy rate? What's the difference between Embryos and Normal Embryos in Table 3? How are the data in tables 4 and 6 counted? There is no standard deviation. Please indicate the number of experiments. <63.4% b (12/22) b> in Table 6 needs to be revised. Please simplify the summary section. Please go deeper into the discussion section.

---

## Round 0.2 · accepted · Accept

Based on the comments of the reviewers and the improved quality of your revised version, your manuscript is ready for acceptation to be published in PeerJ.

# Reviewer 1 ·

Basic reporting

no comment

Experimental design

no comment

Validity of the findings

no comment

Additional comments

It is a good paper to state that the effect of melatonin treatment on the efficiency of super-ovulation and timed artificial insemination in sheep. The positive results shed a light on improvement of quality of embryos and fertility of sheep by melatonin administration.

·

Basic reporting

ok

Experimental design

ok

Validity of the findings

ok

Additional comments

ok